# COVID-19 and Lung Mast Cells: The Kallikrein–Kinin Activation Pathway

**DOI:** 10.3390/ijms23031714

**Published:** 2022-02-02

**Authors:** Seigo Nagashima, Anderson Azevedo Dutra, Mayara Pezzini Arantes, Rafaela Chiuco Zeni, Carolline Konzen Klein, Flávia Centenaro de Oliveira, Giulia Werner Piper, Isadora Drews Brenny, Marcos Roberto Curcio Pereira, Rebecca Benicio Stocco, Ana Paula Camargo Martins, Eduardo Morais de Castro, Caroline Busatta Vaz de Paula, Andréa Novaes Moreno Amaral, Cleber Machado-Souza, Cristina Pellegrino Baena, Lucia Noronha

**Affiliations:** 1Postgraduate Program of Health Sciences, School of Medicine, Pontifícia Universidade Católica do Paraná, Curitiba 80910-215, Brazil; andersonazevedodutra@yahoo.com.br (A.A.D.); mpezziniarantes@gmail.com (M.P.A.); rafaelaczeni@gmail.com (R.C.Z.); carolline.ck@gmail.com (C.K.K.); flavia.centenaro@gmail.com (F.C.d.O.); giulia.piper@gmail.com (G.W.P.); isabrenny@icloud.com (I.D.B.); marcos.curcio@pucpr.edu.br (M.R.C.P.); rebecca.stocco@pucpr.edu.br (R.B.S.); anapaulacamargo@hotmail.com (A.P.C.M.); carolbvaz@gmail.com (C.B.V.d.P.); andrea.moreno@pucpr.br (A.N.M.A.); cbaena01@gmail.com (C.P.B.); 2Postgraduate Program in Biotechnology Applied in Health of Children and Adolescent, Instituto de Pesquisa Pelé Pequeno Príncipe, Faculdades Pequeno Príncipe, Curitiba 80250-060, Brazil; medmoca@gmail.com (E.M.d.C.); cleberius@gmail.com (C.M.-S.); 3Marcelino Champagnat Hospital, Curitiba 80020-110, Brazil

**Keywords:** COVID-19, mast cells (MCs), kallikrein–kinin system (KKS), vascular hyperpermeability, edema, inflammation

## Abstract

Mast cells (MCs) have relevant participation in inflammatory and vascular hyperpermeability events, responsible for the action of the kallikrein–kinin system (KKS), that affect patients inflicted by the severe form of COVID-19. Given a higher number of activated MCs present in COVID-19 patients and their association with vascular hyperpermeability events, we investigated the factors that lead to the activation and degranulation of these cells and their harmful effects on the alveolar septum environment provided by the action of its mediators. Therefore, the pyroptotic processes throughout caspase-1 (CASP-1) and alarmin interleukin-33 (IL-33) secretion were investigated, along with the immunoexpression of angiotensin-converting enzyme 2 (ACE2), bradykinin receptor B1 (B1R) and bradykinin receptor B2 (B2R) on *post-mortem* lung samples from 24 patients affected by COVID-19. The results were compared to 10 patients affected by H1N1pdm09 and 11 control patients. As a result of the inflammatory processes induced by SARS-CoV-2, the activation by immunoglobulin E (IgE) and degranulation of tryptase, as well as Toluidine Blue metachromatic (TB)-stained MCs of the interstitial and perivascular regions of the same groups were also counted. An increased immunoexpression of the tissue biomarkers CASP-1, IL-33, ACE2, B1R and B2R was observed in the alveolar septum of the COVID-19 patients, associated with a higher density of IgE^+^ MCs, tryptase^+^ MCs and TB-stained MCs, in addition to the presence of intra-alveolar edema. These findings suggest the direct correlation of MCs with vascular hyperpermeability, edema and diffuse alveolar damage (DAD) events that affect patients with a severe form of this disease. The role of KKS activation in events involving the exacerbated increase in vascular permeability and its direct link with the conditions that precede intra-alveolar edema, and the consequent DAD, is evidenced. Therapy with drugs that inhibit the activation/degranulation of MCs can prevent the worsening of the prognosis and provide a better outcome for the patient.

## 1. Introduction

The Coronavirus Disease 2019 (COVID-19) pandemic caused by Severe Acute Respiratory Syndrome Coronavirus 2 (SARS-CoV-2) still challenges scientists and health agents worldwide [1]. Up to now, 5,560,718 people have died across the planet due to the disease [2]. By way of comparison, it is estimated that the last pandemic caused by a respiratory virus, Influenza A virus subtype H1N1 (H1N1pdm09), led to the deaths of 123,000 to 395,600 people in the period between 2009 and 2010 [3,4].

When SARS-CoV-2 comes into contact with its host, it binds through the S-spike protein, which has a greater affinity with the receptor-binding domain of the angiotensin-converting enzyme 2 (ACE2), a metalloprotease expressed in the pulmonary and vascular epithelium, oral and gastric mucosa, kidney, heart and brain [5,6]. Interestingly, ACE2 expression is low in the pulmonary epithelium (type I- and II-pneumonocytes) [7]. ACE2 acts on the Renin–Angiotensin–Aldosterone System (RAAS), a chain of enzymatic reactions [8] whose primary function is the maintenance of blood pressure and has a unique role by acting in the degradation of the peptide hormone angiotensin II (Ang II). Ang II causes vasoconstriction, which increases blood pressure and inflammation, and is converted into angiotensin 1–7 (Ang1–7), whose antagonistic effects act on vasodilatation and anti-inflammatory responses [9].

After contact and infection of the pulmonary epithelium, SARS-CoV-2 ends up damaging these tissues and provides the synthesis and secretion of proinflammatory cytokines, such as interleukins (ILs) 1 and 6, and tumor necrosis factor-alpha (TNF-α), among a large range of cytokines responsible for signaling infection by intracellular pathogens, such the interferons (IFNs) α and β, thus contributing strongly to the initiation of the cytokine storm [10]. This overproduction of cytokines increases the local and systemic tissue damage associated with an exacerbated recruitment of inflammatory cells and contributes to diffuse alveolar damage (DAD), apoptosis and pyroptosis in pneumocytes and endothelial cells from the alveolar-capillary pulmonary septum [10,11]. The severity of the injury must be associated with virus infection itself, as well as cytokine storm, making the immune system’s role in the progression and severity of the disease increasingly evident [12,13].

The pyroptotic process consists of an alternative programmed cell death, which activates an inflammasome mediated mainly by caspase-1 (CASP-1) and culminates in the formation of pores over the plasma membrane. The activation of the proteolytic cascade of this enzyme catalyzes the synthesis and secretion of several proinflammatory cytokines, including IL-1β and IL-18 [12,14]. Cells can detect signs of invasion by intracellular pathogens through signaling by membrane receptors, initiating a proteolytic cascade that terminates in their self-destruction [15,16]. Along with its self-destruction, there is a release of proinflammatory cytokines, viral and cellular fragments resulting from cellular destruction, alarmins, Pathogen-Associated Molecular Patterns (PAMPs) and Danger-Associated Molecular Patterns (DAMPs) [16]. Alarmins, associated with several cytokines, act in the sensitization of a class of granular cells that play the role of sentinels in the organism—mast cells (MCs) [17].

MCs are connective tissue-resident cells that present several cytoplasmatic granules, concentrated in regions that directly contact the environment [18]. MCs are known for their action on allergic and inflammatory responses. When antigens trigger their activation through crosslinking with immunoglobulin E (IgE), binding to their FcεRI receptor, these cells start a cascade of chemical granule secretion stored in their cytoplasm [19]. The degranulation process is divided, chronologically, into two distinct stages: (i) immediate, right after its activation, where the secretion of pre-stored mediators in MCs cytoplasm occurs, and (ii) late, or de novo, where the mediators’ secretion occurs hours after their activation. Examples of immediate mediator secretion include vasoactive amines, such as histamine, glycosaminoglycans, such as heparin, proteases, such as tryptase and chymase, and lipid mediators, such as prostaglandins and leukotrienes and the cytokine TNF-α [17,20]. Examples of de novo mediator secretion include cytokines, such as interleukins, chemokines and growth factors [19,21].

As a result of the activation and degranulation of MCs, a large variety of physiological responses are triggered locally and systemically, including the activation of the kallikrein–kinin system (KKS) [22].

Kallikrein is a polypeptide with degradation functions. In KKS, it plays a fundamental role in the proteolytic cleavage of high-molecular-weight kininogens (HMWK), synthesized by the liver, and low-molecular-weight kininogens (LMWK), which are synthesized by tissues [23]. As a result of the action of kallikrein, HMWK is degraded in the hormone bradykinin (BK), while LMWK is metabolized in the form of lys-bradykinin (Lys-BK), with identical physiological action to BK; both target the bradykinin receptor B2 (B2R) [24,25]. Through the action of the Kininase I, BK and lys-BK are cleaved into the des-Arg^9^-BK (DABK) and lys-des-Arg^9^-BK (Lys-DABK), respectively; both target the bradykinin receptor B1 (B1R) [26,27].

While B2R is constitutively present in several healthy tissues, especially in the endothelial vasculature [28], B1R is activated focally in inflammatory events, induced by cytokines such as TNF-α and IL-1β, and also by the proper agonists DABK and lys-DABK [29,30]. The binding of these receptors triggers the regulation of vascular tonus and inflammatory processes, including increased vascular permeability and pain [29]. Interstitial and intra-alveolar edema events, preceded by hyaline membranes and DAD [31], may be related to the increased vascular permeability episodes caused by an upregulation of KKS/BK/B1-2R mediated by lung MCs that are present in the perivascular environment.

Our study investigated the SARS-CoV-2 infection in the lung environment that contributes to the activation of the MCs/KKS with consequent inflammatory processes exacerbation. Avoiding these deregulated inflammatory processes and an exaggerated vascular permeability, whether through therapies with the use of corticosteroids or modulators of MCs, is a challenge that can prevent worsening prognoses and provide a better outcome.

## 2. Results

Demographic data (gender and age), and clinical and histopathological findings can be seen in Table 1, as well as the analyses of the tissue expression of ACE2, IL-33, CASP-1, B1R and B2R. The amounts of IgE^+^ MCs, tryptase^+^ MCs and MCs stained in TB can also be seen in Table 1.

The ACE2 tissue immunoexpression was increased in the COVID-19 group compared to H1N1 and CONTROL groups (*p* = 0.0005 and *p* < 0.0001, respectively). The COVID-19 group presented higher tissue immunoexpression of IL-33 compared to the CONTROL group (*p* = 0.013); however, it was not statistically significant when compared to the H1N1 group. Regarding CASP-1, the COVID-19 group presented higher tissue immunoexpression than both the H1N1 and CONTROL groups (*p* < 0.0001 and *p* < 0.0001, respectively). In terms of the tissue immunoexpression of BK receptors and their cleaved forms, the COVID-19 group had a significant increase in B1R (*p* < 0.0001) and B2R tissue expression (*p* < 0.0001) compared with the CONTROL group. In comparison with the H1N1 group, there was no statistical significance for both receptors (Table 1, Figure 1 and Figure 2).

Respecting the distribution of MCs in the interstitial and perivascular sites, the COVID-19 group presented a higher number of IgE^+^ MCs than the CONTROL group (*p* < 0.0001); however, there was no statistical significance when compared with the H1N1 group. The number of Tryptase^+^ MCs and TB metachromatic MCs was also increased in the COVID-19 group, compared to the H1N1 and CONTROL groups (*p* = 0.039 and *p* < 0.025 vs. *p* = 0.0006 and *p* < 0.0001, respectively). The results can be seen in Table 1 and Figure 1 and Figure 2. The COVID-19 patients were split into two groups regarding the ICU administration of corticosteroids. There was no statistical difference between these two groups of patients related to the number of Tryptase^+^ MCs and TB metachromatic MCs (Table 2).

## 3. Discussion

### 3.1. Pathophysiology of SARS-CoV-2 Infection in the Lower Airways and Mast Cells Activation

MCs play an essential role in the first line of defense against pathogens. Their role is already well elucidated in events related to bacteria and helminths and situations associated with allergies and hypersensitization [17]. Nonetheless, when it comes to infections caused by viral etiologic agents, the role of these connective tissue vigilantes deserves greater attention, mainly related to respiratory viruses that trigger pandemics. Studies have shown an increase in the density of MCs, associated with the damage induced by them, in rodents infected with H5N1 Influenza virus [32], although their role in this infection is not entirely clear. In many infectious processes caused by viruses, MCs have a protective effect; however, in infectious conditions caused by viruses of greater pathogenicity, such as Influenza itself, their role in regulating homeostasis becomes questionable [33,34].

Recent studies indicate that SARS-CoV-2 activates innate immunity cells, including MCs, favoring the unregulated increase in cytokine storm conditions [35,36]. The increased presence of MCs in lung tissue from patients who died from the severe form of the disease is remarkable, as previously noted by Motta Jr et al. [37]. Another study identified many activated MCs in the bronchoalveolar lavage of patients affected by COVID-19 [38]. The IgE-mediated activation process is one of the factors (but not the only one) promoting the release of mediators. MCs conserve a myriad of different classes of membrane receptors that, when signaled through their ligands, also induce the degranulation of this sentinel cell [39]. Among the receptors that do not depend on IgE for MC activation are the Pattern-Recognition Receptors (PRRs), responsible for recognizing PAMPs and DAMPs, alarmins and cytokines [16,39].

In our study, we observed a large number of IgE^+^ MCs in patients who died from COVID-19 compared to the CONTROL group (Table 1, Figure 1) but not when compared to the H1N1 group. Graham et al. demonstrated that MCs have essential participation in inflammatory events caused by the H1N1 Influenza A virus [40]. This large number of IgE^+^ MCs present in pandemic groups corroborates previous studies.

Pyroptosis is notably present in the infection processes of cells parasitized by SARS-CoV-2. A recent study demonstrated pyroptotic processes and a CASP-1-mediated inflammasome formation in human monocytes infected with SARS-CoV-2 [41]. The activation of this proteolytic cascade, mediated mainly by CASP-1, results in the synthesis and secretion of several proinflammatory cytokines, in addition to the release of alarmins, PAMPs and DAMPs, which have an affinity for PRRs [42]. PRR, best understood by toll-like receptors, including TLR3, TLR7 and TLR8, are constitutively present in MCs and can recognize viral RNA [39,43]. Activation of MCs via TLR3 is caused by viral stimuli and may contribute to the recognition of endogenous danger signals. External signals such as mRNA originating from necrotic cells also stimulate activation via TLR3 [44].

Through the higher values of CASP-1 tissue immunoexpression in COVID-19 patients, we could demonstrate the evidence of pyroptosis, leading to a possible induction of activation and degranulation of MCs (Figure 3).

Interleukin-33 receptor (IL-33R), a protein of the IL-1 family, widely expressed on the MCs’ surface, plays an essential role in the MCs’ activation. IL-33, a type-1 cytokine also belonging to the IL-1 family, is immunoexpressed mainly in epithelial cells that directly contact the external environment. IL-33/IL-33R are activated after injury insults and inflammation and have an alarmin function [39,45]. In our findings, we identified a larger tissue expression in this cytokine regarding COVID-19 patients when compared to the CONTROL group. These findings corroborate the literature; once SARS-CoV-2 inflammation is associated with COVID-19, higher immunoexpression of IL-33 also therefore contributes to the MCs’ activation. Studies show that Herpes simplex virus 2-infected epithelial cells that secreted IL-33 induced MCs to secrete IL-6 and TNF-α without their whole degranulation [46].

### 3.2. Mast Cell Degranulation and the Role of Its Mediators in the Kallikrein–Kinin System

Metachromasia is a striking feature in the impregnation of MCs when stained with TB. Its cytoplasmatic granules, rich in heparin, heparan sulfate and proteoglycans, among other polyanions, are related to its dense negative charge, which polymerizes the dye molecules, highlighting these structures [47,48]. The staining of MCs with TB is a very efficient method for studying their morphology and functions based on their stored mediators. MCs provided with cytoplasmatic granules can be contemplated by metachromatic differentiation [48].

Our study, as well as that of Motta Jr et al., identified a higher number of metachromatic MCs carried with cytoplasmatic granules or in the process of degranulation in patients who died from COVID-19 (Figure 1).

Heparin also plays an essential role in the maturation and stabilization of another cytoplasmatic mediator—tryptase [49]. Tryptase, a trypsin-like serine protease secreted by MCs and other granulocytes, is involved in allergic reactions [50]. Tryptase plays an essential role in the distribution and activation of MCs [51]. Its physiological functions, together with histamine, are important in vascular hyperpermeability, and in interstitial and intra-alveolar edema. Interestingly, this enzyme is also involved in tissue remodeling [52,53]. Given this, tryptase is a protease whose secretion may indicate MC activation [51,54]. In addition, tryptase can also induce nearby MCs to degranulate their mediators, contributing to the amplification of this signal [55,56].

Our study showed an increased number of activated Tryptase-secreting MCs in patients affected by COVID-19 compared to the other groups (Figure 1), thus correlating these cells with the events of acute lung injury, DAD and terminal alveolar fibrosis.

Heparin indirectly appears as a pivot to initiate a complex inflammatory response involving the Contact Activation System (CAS), proceeding in a proteolytic cascade acting on intravascular coagulation, inflammation and vascular permeability [24,39,57]. Heparin leads to the proteolytic cleavage of Hageman’s factor (FXII) to its activated form (FXIIa) [58]. FXIIa plays a central role in CAS, whether acting directly at the start of the coagulation cascade or activating the kallikrein–kinin system (KKS) by converting prekallikrein into its active circulating form—kallikrein. Kallikrein, in its turn, can hydrolyze the neutral form of FXII, triggering a positive feedback loop [23,25,58].

Not only does heparin and its contact with other negatively charged surfaces trigger the activation of FXII/FXIIa. Collagen is also a molecule that provides, due to its anionic nature, the activation of FXII/FXIIa. Collagen is immensely present in the basement membrane of the subendothelial region. As a result of vascular injury and endothelial cell death caused by the SARS-CoV-2 infection or cytokine storm, the subendothelial collagen is exposed [14,59]. Thrombotic events [60] are also described in the COVID-19 and may be related to FXII/FXIIa activation and coagulation cascade modulation [61].

FXIIa also activates KKS, where kallikrein plays a crucial role in the proteolytic cleavage of HMWK into BK and LMWK into lys-BK, both having B2R affinity, resulting in increased vascular permeability [23,62,63]. BK and lys-BK are also cleaved [26], forming DABK and lys-DABK, respectively, which have an affinity for B1R and contribute to increasing the vascular permeability [63].

In our study, both tissue immunoexpression of B2R and B1R were increased in patients affected by COVID-19 compared to the CONTROL group. The exacerbation of pulmonary vascular permeability and plasma extravasation towards the pulmonary interstitium to the alveolar lumen was evident once intra-alveolar edema was also observed in patients associated with the COVID-19 and H1N1 groups, due to the presence of activated bradykinin receptors (Figure 2). These aspects can directly link the plasma leakage events to the KKS activation (Figure 4).

As already described, SARS-CoV-2 adopts the ACE2 receptor to invade the host cell [64]. A significant amount of ACE2 is lost during viral infection when the viral S-spike subunits bind with ACE2 receptor intervenes in the synthesis of a protein complex. A Transmembrane Serine Protease 2 (TMPRSS2) and furin, constitutively present in the plasma membrane of host cells, cleave the S-spike subunits into S1 and S2, allowing membrane fusion and insertion of viral RNA [65], causing a loss of ACE2 function. In an eventual absence of these surface enzymes, the virion particle can infect the host cell via endocytosis, along with ACE2 itself [66,67].

In addition to its essential role in RAAS, downregulation of ACE2, which can degrade DABK and lys-DABK, promotes the increasing of these hormones that, by binding B1R [65], can trigger the increase in vascular permeability and nitric oxide (NO) and cytokines’ secretion [68,69,70].

Evidence pointed out by the *B1R* gene expression showed that B1R expression is increased during hypertension events [27]. This risk factor may be directly associated with augmented B1R and worsening of the disease. Sodhi et al. demonstrated that the loss of pulmonary ACE2 culminated in the activation of DABK-lys-DABK/B1R and the secretion of proinflammatory cytokines such as CXCC5 and TNF-alpha in alveolar epithelium of rats [68].

In the infectious event of COVID-19, a downregulation of ACE2 can also contribute to cytokine storms. Interestingly, our findings demonstrated an increase in tissue ACE2 in patients in the COVID-19 group compared to patients in other groups, which is corroborated by the literature [71]. This eventual upregulation would be due to the association between the advanced age and the use of mechanical ventilation, supported by a study designed by Baker et al., where the increasing of ACE2 was demonstrated by its higher levels of gene expression and the enzyme immunoexpression in the alveolar epithelium in these patients [72]. In addition, the number of days of mechanical ventilation in COVID-19 patients was longer than H1N1. On the other hand, Wang et al., in an in vitro study, identified the recycling of ACE2 back to the plasma membrane of 293E-ACE2-GFP lineage cells, which occurred 14 h after contact and endocytosis promoted by the S-spike protein [73]. This increased amount of tissue ACE2 in our COVID-19 patients was insufficient to suppress a DABK-lys-DABK/B1R activation.

From the point of view of Nicolau et al., a loss of ACE2 causes triple harm to the patient: (i) increased levels of Ang II, (ii) decreased levels of Ang 1–7, and (iii) increased activation of DABK-lys-DABK/B1R [74]. Thus, a possible therapy would be the use of soluble ACE2 in order to trap the virus and inactivate it, as proposed by Alhenc-Gelas and Drueke, combined with the positive effects carried by the direct action of ACE2 on RAAS and KKS [75,76]. Among the characteristic histopathological findings of the disease, intra-alveolar edema is present, as mentioned, resulting from the invasion of plasma exudate carrying coagulation factors and molecules of the complement system arising from vascular hyperpermeability. Hyperpermeability was observed by Garvin et al., who linked it with an excess of bradykinin. In other words, a bradykinin storm affects some patients with the disease [77]. Our descriptions also include the presence of intra-alveolar edema in patients who comprise the three groups. Edema was not higher in patients affected by COVID-19 than in patients affected by H1N1; however, there was a statistically significant difference compared to patients in the CONTROL group.

Corticosteroid therapy may also promise to avoid MCs’ action in the inflammatory context, as well as exaggerated vascular permeability. It was previously shown that hydrocortisone and dexamethasone inhibit MCs’ degranulation process [78]. Corticosteroids reduce the synthesis and secretion of IL-3, a fundamental cytokine for the maturation and recruitment of several hematopoietic cell lines, including MCs [79]. The absence of this cytokine still promotes MCs’ apoptosis [80]. Adverse effects with prolonged use, even at low doses, range from skeletal muscle, endocrine and metabolic, cardiovascular and dermatological dysfunctions to immunological side effects [81,82].

However, the ICU corticosteroid therapy may not attenuate MCs’ activation and degranulation process in patients affected by COVID-19. This fact may be supported by our results, where those patients who received corticosteroid therapy had no difference from those that did not receive these drugs in terms of the number of activated MCs. One justification would be the time the drug was administered, as the protocols and guidelines adopted by the Marcelino Champagnat Hospital are being phased out by the World Health Organization, the European Medicines Agency, the UK Chief Medical Officer and the US National Institutes of Health, who recommend the initiation of corticosteroid therapy in patients who are already hospitalized and at the moment of oxygen therapy becoming required, regardless of mechanical support [83,84,85].

Stabilizers are an alternative to avoid the problems caused by cytoplasmatic mediators secreted by MCs. Luteolin is a polyphenolic flavonoid and, as with its analog and innovative tetramethoxyluteolin, inhibits MCs’ degranulation by blocking intracellular calcium channels related to mediator secretion processes. These modulators also inhibit the activation of NF-κB [86,87], a nuclear factor responsible for the transcription of several cytokines secreted by MCs and related to cytokine storm [88]. Modulator-based therapy may promise to inhibit the action of MCs on COVID-19 and, thus, prevent the worsening of the clinical conditions of patients.

Formalin-fixed and paraffin-embedded (FFPE) can be considered the most significant limitation in our study since this material provided only static information regarding the moment of the patient’s death. There was no possibility of retrieving information during the disease processes. Additionally, our sample is small, given the small number of *post-mortem* lung samples due to the difficulty in collecting them in this highly contagious COVID-19 ICU environment. Although the results are consistent, a larger cohort study would provide a more effective investigation of the data and more accurate results. This work can be considered a preliminary study regarding the small number of samples, suggesting a larger cohort for future investigations.

## 4. Materials and Methods

### 4.1. Post-Mortem Samples and Histochemistry Assay

This study was approved by the National Research Ethics Committee (Comitê Nacional de Ética em Pesquisa—CONEP), under the numbers 3.944.734/2020 (COVID-19 patients) and 2.550.445/2018 (H1N1 and CONTROL patients). The methodology was performed following relevant guidelines and regulations. The patients’ legal representatives allowed the use of the lung samples for this study.

The clinical data of patients who died of COVID-19 (COVID-19 group, *n* = 24) and H1N1pdm09 (H1N1 group, *n* = 10) came from the medical records of the Intensive Care Unit (ICU) of the Marcelino Champagnat Hospital and the ICU of Hospital de Clínicas, in Curitiba, Brazil, respectively. The *post-mortem* lung samples, measuring 3 × 3 cm, were obtained through left anterior mini-thoracotomy with upper left lobe segment resection. Testing for the SARS-CoV-2 and H1N1pdm09 viruses was performed on nasopharyngeal smears (RT-qPCR).

The clinical data of patients who died from cardiovascular and neoplastic diseases not related to lung lesions (CONTROL group, *n* = 11) were taken from the medical records of the Hospital de Clínicas. The *post-mortem* lung samples were sourced from the hospital’s necropsy sample bank.

The *post-mortem* lung samples were FFPE to proceed with the Hematoxylin-Eosin (HE) staining and histopathological characterizations. From FFPE lung samples, multi-sample paraffin tissue blocks (Tissue Microarray, TMA) were constructed to facilitate the execution of the histochemical and immunohistochemical assays. Three cylindrical fragments (chosen from damaged areas using HE staining), measuring 0.3 cm in diameter, were removed from the original blocks (donors FFPE blocks) and organized into new multi-sample paraffin blocks (recipient blocks).

Intact and degranulation MCs were identified by means of the Toluidine Blue (TB) histochemical technique (Scientific Exodus, cod. AT9115SO, São Paulo, Brazil) due to its metachromatic characteristic. These MCs were observed exclusively in the alveolar septum and perivascular spaces by counting stained cells in 20 high-power fields (HPF, 40X, Olympus Objective, 0.26 mm^2^ per patient), using a BX50 optical microscope (OLYMPUS, Tokyo, Japan). Average scores were obtained by screening 20 randomized HPFs (total area of 5.2 mm^2^ per case). Data were subjected to statistical analysis.

### 4.2. Immunohistochemistry Assay

As read-outs, an immunohistochemical technique was performed in order to identify the immunoexpression of IL-33 (anti-IL-33, Polyclonal/rabbit, code A8096, 1:800 dilution, ABclonal, Manhattan Beach, CA, USA), B1R (anti-B1R, Polyclonal/rabbit, GTX70845, 1:100, GeneTex, Irvine, CA, USA), B2R (anti-B2R, Polyclonal/rabbit, ab236093, 1:100, Abcam, Cambridge, UK), CASP-1 (anti-CASP-1, Polyclonal/rabbit, ab189796, 1:200, Abcam, Cambridge, UK) and ACE2 (anti-ACE2, Polyclonal/rabbit, ab272690, 1:50, Abcam, Cambridge, UK) for observation of its immunoexpression in alveolar macrophages, endothelial cells and, type-I and -II pneumocytes. Tissue immunoexpression of Immunoglobulin (Ig) E (anti-IgE, Polyclonal/rabbit, BSB3070, 1:100, Bio SB, Santa Barbara, CA, USA) was used to quantify IgE+ MCs. Immunoexpression of tryptase (anti-Tryptase, Monoclonal/rabbit, EP259, 1:400, BioSB, Santa Barbara, CA, USA) was used to identify activated Tryptase+ MCs, as well as MCs in the process of degranulation of this enzyme.

The secondary polymer was the multipurpose developer’s Mouse and Rabbit Specific HRP/DAB IHC Detection Kit - Micro-polymer, ab236466 (Abcam, Cambridge, UK). Specificity controls were performed by (i) omitting the primary antibody (negative control) and (ii) conducting a tissue sample test on positive controls for each immune marker.

### 4.3. Morphometric Analysis and MC-Counting Process

The slides immunolabeled with anti-IL-33, anti-CASP-1, anti-B1R, anti-B2R and anti-ACE2 were scanned with the assistance of the Axio Scan Z1 slide scanner (Zeiss, Jena, Germany) and submitted to the generation of 30 HPF (COVID-19 group) and 20 HPF (H1N1 and CONTROL groups) by the ZEN Blue Edition software (Zeiss, Jena, Germany). Analyses were performed blindly by an observer. The areas of immunoexpression were quantified using Image Pro-Plus 4.5 software (Media Cybernetics, Rockville, MD, USA), and subsequently, these areas were converted to percentages. Data were subjected to statistical analysis.

The slides immunolabeled with anti-IgE and anti-Tryptase were observed exclusively in the alveolar septum and perivascular spaces by counting immunostained MCs in 20 randomized HPF (40X, Olympus Objective, 0.26 mm^2^ per sample), using a BX50 optical microscope (OLYMPUS, Tokyo, Japan). Average scores were obtained by screening 20 randomized HPFs (total area of 5.2 mm^2^ per case). Data were subjected to statistical analysis.

### 4.4. Statistical Analysis

The normality condition was evaluated using the Shapiro–Wilk test. The nonparametric test for the continuous variables was performed using the Mann–Whitney test, with the values characterized by the median, interquartile range, minimum and maximum values. The results of the parametric test for the continuous demographic and clinical variables between two groups, performed using the Student’s *t*-test, were characterized by the mean and straight deviation values. For categoric variables, the performed test was Fisher’s exact test, and its values were characterized by frequency. Values of *p* < 0.05 indicated statistical significance. The data were analyzed using the software JMP (™) Pro 14.0.0. (SAS Institute, Cary, NC, USA).

## 5. Conclusions

In conclusion, the direct participation of MCs in inflammatory events related to vascular hyperpermeability resulting from KKS activation is evident. The use of drugs that inhibit the degranulation of MCs, and therefore inhibit the action of KKS, is a valid option to prevent the disease from worsening. Corticosteroids are efficient in reducing the number of MCs. However, their effectiveness in preventing severe clinical outcomes depends on the moment they are administered. Under prolonged use, corticosteroids may have adverse effects on the patient. As an alternative, the use of MC inhibitors appears as an option to prevent the adverse effects brought about by the continuous use of corticosteroids.

## Figures and Tables

**Figure 1 ijms-23-01714-f001:**
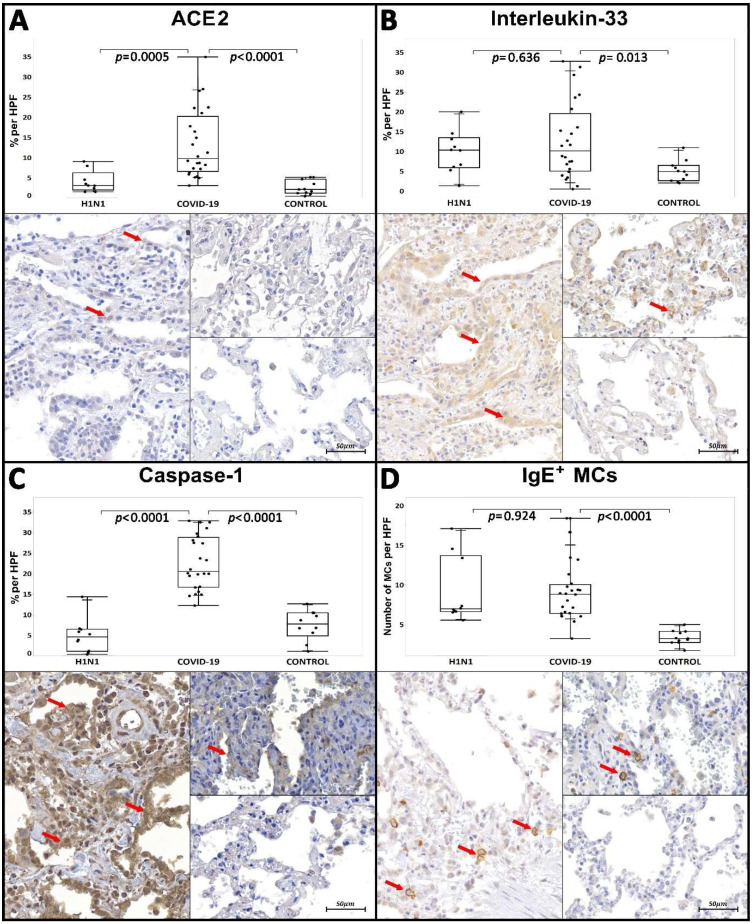
Graphics show comparison between COVID-19 (left images), H1N1 (right upper images) and CONTROL (right lower images), regarding ACE2, Interleukin-33 and Caspase-1 in the percentage of immunoexpression per HPF (high power field) and the density of IgE^+^ MCs per HPF. (**A**) ACE2 showed mild positive immunostaining (red arrow) in the alveolar epithelium of the COVID-19 group. However, there was a significant statistical increase compared to the H1N1 and CONTROL groups. (**B**) Tissue expression of Interleukin-33 (red arrow) appeared higher in patients in the COVID-19 group when compared to the CONTROL group; however, it was not statistically increased compared to the H1N1 group. (**C**) The tissue expression of Caspase-1 (red arrow) was increased in the COVID-19 group when compared to the H1N1 and CONTROL groups, demonstrating a larger significant presence of pyroptosis by SARS-CoV-2 infected cells. (**D**) Regarding the density of IgE^+^ MCs, the COVID-19 group presented an increase in the number of MCs opsonized by IgE (cells highlighted by red arrows) when compared to the CONTROL group. However, there was no statistical difference compared to the H1N1 group. Images were scanned using the Axio Scan Z1 slide scanner (Zeiss, Jena, Germany) at 40× magnification. Mann–Whitney nonparametric test. Values of *p* < 0.05 indicated statistical significance.

**Figure 2 ijms-23-01714-f002:**
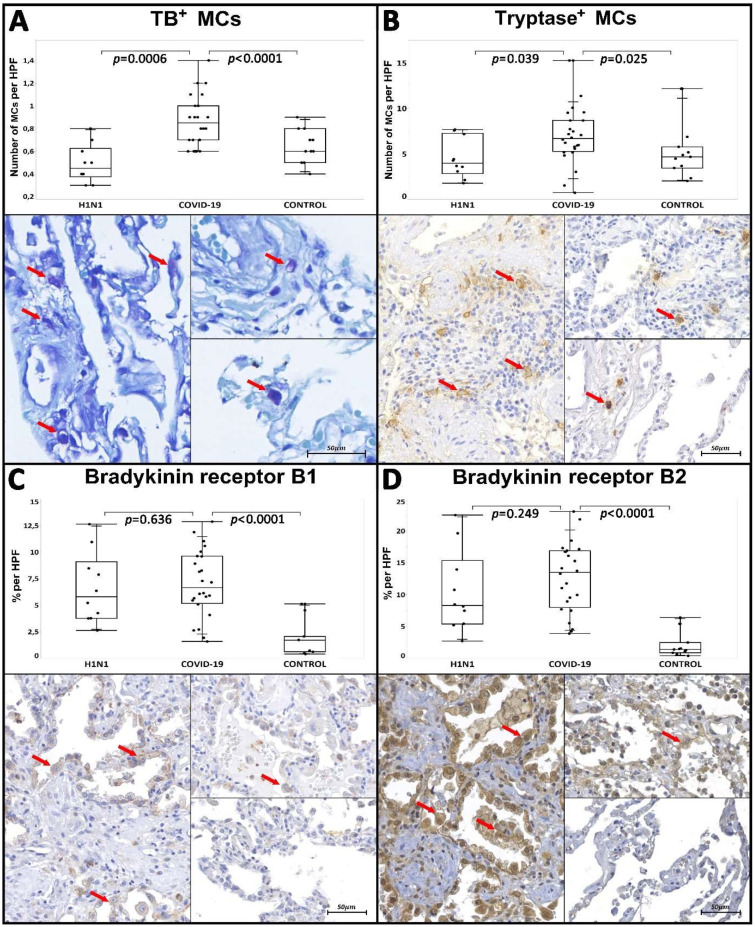
Graphics show Comparison between COVID-19 (left images), H1N1 (right upper images) and CONTROL (right lower images), regarding the density of Toluidine Blue (TB) metachromatic MCs, Tryptase^+^ MCs per HPF (high power field) and tissue Bradykinin receptor B1 and Bradykinin receptor B2 in the percentage of immunoexpression per HPF. (**A**) Metachromatic MCs stained with TB are shown (red arrows) at higher density in the COVID-19 group when compared to the H1N1 and CONTROL groups. (**B**) The density of Tryptase^+^ MCs (red arrow) was also increased in the COVID-19 group when compared to the H1N1 and CONTROL groups. (**C**,**D**) As for tissue immunostaining of Bradykinin receptors, B1R and B2R was higher in the COVID-19 group (highlighted by red arrows) in comparison to the CONTROL group. However, there was no statistically related difference with the H1N1 group. Images were scanned using the Axio Scan Z1 slide scanner (Carl Zeiss, Jena, Germany): (**B**–**D**) at 40× magnification and (**A**) at 63× magnification. Mann–Whitney nonparametric test. Values of *p* < 0.05 indicated statistical significance.

**Figure 3 ijms-23-01714-f003:**
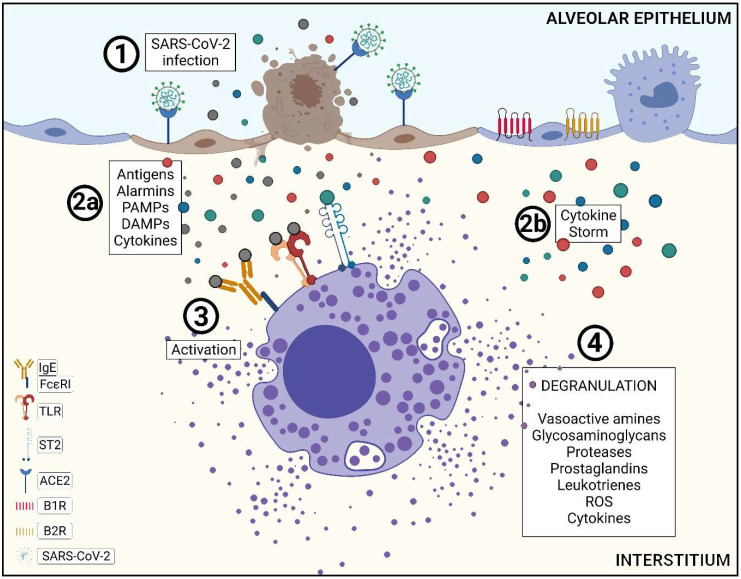
Pathophysiology of SARS-CoV-2 infection in the lower airways and mast cell activation. (1) With the arrival of SARS-CoV-2 in the alveolar environment, it starts to infect type-I and -II pneumocytes, which express ACE2 in their plasma membrane. ACE2 has an affinity with the viral S-Spike protein, used to enter the host cell. (2a) The infectious process causes pneumocytes to die by various means, especially pyroptosis, and this is mainly mediated by CASP-1. The death process by pyroptosis leads to the secretion of antigens, alarmins, PAMPs, DAMPs and cytokines. IL-33, which has the ST2 receptor, acts as an alarmin and is secreted by cells affected by inflammatory insults, present in the infectious processes caused by SARS-CoV-2. (2b) This significant cytokine secretion contributes to the cytokine storm. (3) A large amount of pyroptotic elements triggers the recruitment and activation of alveolar interstitium-resident mast cells (MCs) through the binding of antigens and alarmins with several sensitive receptors present in their plasma membranes. Fc epsilon RI (FcεRI) has an affinity for IgE sensitized by antigens, and toll-like receptors (TLRs) also have an affinity for antigens and alarmins. (4) As a result of activation, MCs secrete many classes of chemical mediators that act on inflammatory processes. Examples are vasoactive amines, glycosaminoglycans, proteases, lipid mediators, reactive oxygen species (ROS) and a large class of cytokines, which play an essential role in inflammatory events and in the recruitment of other inflammatory cells.

**Figure 4 ijms-23-01714-f004:**
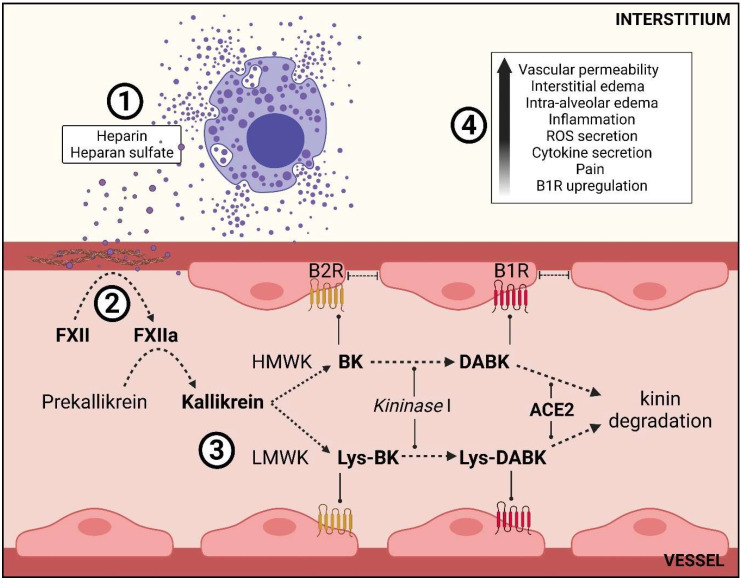
Mast cell degranulation and the role of its mediators in the kallikrein–kinin system (KKS). (1) Chemical mediators with anionic properties secreted by activated mast cells (MCs), such as heparin and heparan sulfate, (2) together with exposure of collagen from the subendothelial layer of the adjacent vasculature, due to vasculitis caused by SARS-CoV-2 itself or cytokine storm, promote the CAS, in which the conversion of FXII into its activated form (FXIIa) is triggered. (3) KKS initiation: FXIIa converts prekallikrein into its active form, kallikrein, initiating the KKS. Kallikrein converts HMWK and LWMK into BK and lys-BK, respectively. Both hormones interact with the B2R, constitutively expressed in healthy tissues. Kininase I cleave BK and lys-BK into DABK and lys-DABK, respectively. Both new hormones interact with B1R, activated in a manner that is focally influenced by inflammatory events. Interestingly, DABK and Lys-DABK are degraded by ACE2, losing their function. (4) The binding of BK and lys-BK/B2R, and of DABK and lys-DABK/B1R, contribute to inflammatory and hyperpermeability events associated with edema, ROS secretion, pain, cytokine storm and B1R upregulation’s positive loop.

**Table 1 ijms-23-01714-t001:** Comparison between COVID-19, H1N1 and CONTROL groups according to demographic, clinical, histopathological and immunohistochemical findings.

Characteristics		H1N1		COVID-19	Control	
		Values	*p*-Value ^1^	Values	*p*-Value ^2^	Values
Gender ^a.^	Male	8 (80%)	0.437 **	15 (62.5%)	0.709 **	8 (72.7%)
	Female	2 (20%)		9 (37.5%)		3 (27.3%)
Age ^b^		43.5 ± 14	0.001 ***	72 ± 12.5	0.001 ***	42.3 ± 14.3
Time from HospitalizationTo Death ^b^		4.7 ± 6.13	0.003 ***	15.9 ± 10.2	0.051 ***	7.64 ± 13.1
Time of MechanicalVentilation ^b^		4.7 ± 6.13	0.028 ***	12 ± 9.20	-	-
Alveolar Edema ^a^	Present	9 (90%)	0.644 **	19 (79.2%)	0.022 **	4 (36.4%)
	Absent	1 (10%)		5 (20.8%)		7 (63.6%)
Tissue Immunoexpression of ACE2 ^c^		3.07/4.26 (1.51–9.03)	0.0005 *	9.77/13.66 (3.1–34.95)	<0.0001 *	2.14/3.46 (0.46–5.16)
Tissue Immunoexpression of IL-33 ^c^		10.34/7.51 (1.36–19.45)	0.636 *	10.14/14.51 (0.52–32.75)	0.013 *	4.93/3.82 (2.05–10.94)
Tissue Immunoexpression of CASP-1 ^c^		4.57/5,37 (0.36–14.34)	<0.0001 *	20.56/12.1 (12.23–32.85)	<0.0001 *	7.73/5.61 (1.09–12.67)
Tissue Immunoexpression of B1R ^c^		5.79/7.34 (2.62–12.64)	0.636 *	6.65/4.45 (1.58–12.88)	<0.0001 *	1.69/1.65 (0.39–5.11)
Tissue Immunoexpression of B2R ^c^		8.29/10 (2.71–22.49)	0.249 *	13.51/8.87 (3.88–23.06)	<0.0001 *	1.39/1.62 (0.38–6.4)
Number of IgE^+^ MCs ^d^		6.97/7.06 (5.55–17.1)	0.924 *	8.85/3.65 (3.25–18.4)	<0.0001 *	3.25/1.4 (1.75–5)
Number of Tryptase^+^ MCs ^d^		3.95/4.44 (1.75–7.65)	0.039 *	6.65/3.4 (0.7–15.25)	0.025 *	4.65/2.33 (2–12.15)
Number of TB Metachromatic MCs ^d^		0.45/0.25 (0.3–0.8)	0.0006 *	0.85/0.3 (0.6–1.4)	<0.0001 *	0.6/0.3 (0.4–0.9)

Continuous variables expressed as median/interquartile range (minimum–maximum). ^a^ Categorical variables expressed as absolute number (frequency). ^b^ Demographic variable AGE (in years) and clinical variables TIME FROM HOSPITALIZATION TO DEATH (in days) and TIME OF MECHANICAL VENTILATION (in days) expressed as mean ± straight deviation. ^c^ Tissue expression in percentage per HPF. ^d^ Number of cells per 20 HPF (average). ^1^ *p*-values compared between COVID-19 and H1N1 groups. ^2^ *p*-values compared between COVID-19 and CONTROL groups. * Mann–Whitney test. ** Fisher’s exact test. *** Student’s *t*-test.

**Table 2 ijms-23-01714-t002:** Comparison between COVID-19 patients treated and non-treated with corticosteroids, according to the number of activated MCs (Tryptase^+^) and degranulating MCs (Toluidine Blue).

Characteristics	Number of Tryptase^+^ Mcs ^2^		Number of Toluidine Blue Metachromatic Mcs ^2^	
	Values	*p*-Value ^a^	Values	*p*-Value ^b^
Treated 17/24 (70.8%) ^1^	6.2/2.7 (3–11.35)	0.633 *	1.2/0.77 (0.35–2.45)	0.774 *
Non-Treated 7/24 (29.1%) ^1^	7.7/8.05 (0.7–15.25)		1/1.3 (0.55–2.75)	
Corticosterids Administered ^3,4^	Dexamethasone 6 mg/day 12/17 Hydrocortisone 100 mg/day 1/17 Hydrocortisone 200 mg/day 3/17 Methylprednisolone 125 mg/day 2/17 Prednisone 60 mg/day 1/17 Prednisone 10 mg/day 1/17			

Continuous variable by median/interquartile range (minimum–maximum). ^1^ Absolute number (frequency). ^2^ Number of cells per 20 HPF (average). ^a^ *p*-value compared between TREATED and NON-TREATED patients regarding NUMBER OF TRYPTASE^+^ MCs. ^b^ *p*-value compared between Treated and Non-Treated patients regarding NUMBER OF TOLUIDINE BLUE MCs. * Mann–Whitney test. ^3^ Some patients had an association with more than one type of drug. ^4^ There are no data related to H1N1 and CONTROL group patients.

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
