# Peer review of "COVID-19 and Lung Mast Cells: The Kallikrein–Kinin Activation Pathway"

_ijms, 2022, doi:10.3390/ijms23031714_

Round 1

Reviewer 1 Report

Dear Authors,

Thank you for submitting the manuscript titled “COVID-19 and Lung Mast Cells: The Kallikrein-Kinin Activation Pathway”.

The manuscript has written with nice introduction and discussion section. So well done. The data presented in the manuscript are for 10 SARS-CoV-2 infected patients and its correlation with activation of Kallikrein-Kinnin System. The histology study compares H1N1 vs COVID-19 vs Control group for ACE-2, IL-3, CASP-1, BK receptors, number of Tryptase MCs. The presented data clearly indicates the significant presence of above groups in COVID-19 patients and signifies the link of KKS in the COVID-19 patients and deregulation of MCs.

The study is needs more evidence to justify the outcomes and authors are suggested to address below comments.

Major:

  1. The results needs validation by cellular study in Mast cells. Please investigate by ELISA to show the certain cytokines storm after triggering the infection.
  2. Another suggestion will be seeing the cytoplasmic and genomic effect by mRNA or WB results.
  3. Also, how proliferation assay with mast cells to justify the level of inflammation and possible effect of suggested treatment option in manuscript can be tested.
  4. Down stream effect of KKS path and BK can be explored with protein data.

Minor:

  1. Some type error has spotted in manuscript, please proof read it.
  2. Though this is small study, the results are significance, can authors please summaries the limitation part again and includes how bigger cohort study can steer the results and outcomes.

Author Response

Response to the Reviewer:

Manuscript ID: ijms-1552886

Title: COVID-19 and Lung Mast Cells: The Kallikrein-Kinin Activation Pathway

Submitted to the section: Molecular Pathology, Diagnostics, and Therapeutics,

Molecular Advances in SARS-CoV-2 Transmission, Infection and Pathology

We thank the Reviewer and the Editors for the valuable suggestions and the opportunity to improve our manuscript.

  1. The results need validation by cellular study in Mast cells. Please investigate by ELISA to show the certain cytokines storm after triggering the infection.

Thank you for your suggestions. The presence of mast cells is validated by Toluidine Blue metachromatic histological/histochemical staining technique. TB staining is commonly used to highlight cytoplasmatic granules due to its negative charge, polymerizing the dye molecules. Our laboratory has expertise with this proven technique, as observed in this previously published work:

DOI: https://doi.org/10.3389/fimmu.2020.574862

The ELISA technique was not used because our study was performed in formalin-fixed paraffin-embedded samples (FFPE). Cytokine storm has been shown previously in other studies from our research group, as observed in this previously published work:

DOI: 10.1016/j.rmcr.2020.101292

  1. Another suggestion will be seeing the cytoplasmic and genomic effect by mRNA or WB results.

Thank you for the suggestion. Indeed, we agree that some molecular biology techniques could be improved to evaluate tissue samples. However, as presented in the study, the tests were performed in formalin-fixed paraffin-embedded samples (FFPE). FFPE samples are still a significant challenge in studies involving techniques that require RNA extraction or protein assays, such as gene expression and WB studies, due to the degradation of RNA and proteins during processing. For this reason, we chose the immunohistochemistry technique that is most suitable for FFPE samples.

  1. Also, how proliferation assay with mast cells to justify the level of inflammation and possible effect of suggested treatment option in manuscript can be tested.

The use of MC modulators was a mere suggestion in the face of the failure of treatments using corticosteroids. A suggestion would be the association of MC modulators protocols with other drugs that have been already tested.

  1. Downstream effect of KKS path and BK can be explored with protein data.

Indeed, certainly could, but unfortunately, there was no possibility once the proteins involved in the KKS proteolytic cascade are serological and our tests were performed in formalin-fixed paraffin-embedded samples (FFPE). However, the entire process and its outcome could be verified by the positive immunoreactivity of bradykinin 1 and 2 receptors, combined with the events of vascular hyperpermeability and alveolar edema in the patients (19 of the 24 patients) that make up the COVID-19 group (Table 1).

  1. Some type error has spotted in manuscript, please proof read it.

Thank you for the observation. Corrections have been provided.

  1. Though this is small study, the results are significance, can authors please summaries the limitation part again and includes how bigger cohort study can steer the results and outcomes.

A larger cohort surely would improve our results. (lines 419-424).

Additionally, our sample is small, given the small number of post-mortem lung samples due to the difficulty in collecting them in this highly contagious COVID-19 ICU environment. Although the results are consistent, a larger cohort study would provide a more effective investigation of the data and more accurate results. This work can be considered a preliminary study regarding the small number of samples, suggesting a larger cohort for future investigations.

Reviewer 2 Report

To Editor,

The study by Nagashima et al demonstrates the association of Kallikrein-Kinin (KK) pathway in lung mast cells for Covid patients. They examined factors that caused activation and deregulation of mast cells and their adverse effects on the alveolar septum through the release of inflammatory mediators.  Based on this, the authors investigated CASP-1 and alarmin IL-33 together with immunoexpression of ACE-2, B1R and B2R on Covid 19 affected lung samples. Mast cells were further counted. They found an increased immunoexpression of tissue biomarkers CASP-1, IL-33, ACE-2, B1 and B2R in alveolar septum of Covid patients associated with higher density of Mast cells in addition to edema. Based on these results, the authors suggest a direct correlation of MC with vascular hyperpermeability, edema and alveolar damage which possible affects patients from severe Covid. The authors eventually suggest that KK activation is responsible for the exacerbated increase in vascular permeability and possible consequences. The conclude that drugs that inhibit the activation /degranulation of Mast cells can prevent severity and better results for the patient.

The manuscript is well written, however my concerns are outlined below. Spelling mistakes should be corrected throughout the manuscript.

  1. page 5, line 120, presented higher tissue should be described as a trend.
  2. Fig 1, the authors can consider revising plots, control, H1N1 and Covid to facilitate easy reading
  3. No data about medication of these patients were mentioned.
  4. The discussion can be shortened. The authors must discuss the results. It looks like  introduction.
  5. It is known that Covid increases the inflammatory parameters but how mast cells are associated with vascular hyperpermeability is not clearly indicated. 
  6. Furthermore, role of KKS pathway should be linked clearly to the results and discussion.
  7. The authors might consider elaborating the results as well.
  8. The authors have compared the data to control and H1N1. But no discussion was provided at all.

Author Response

Response to the Reviewer:

Manuscript ID: ijms-1552886

Title: COVID-19 and Lung Mast Cells: The Kallikrein-Kinin Activation Pathway

Submitted to the section: Molecular Pathology, Diagnostics, and Therapeutics,

Molecular Advances in SARS-CoV-2 Transmission, Infection and Pathology

We thank the Reviewer and the Editors for the valuable suggestions and the opportunity to improve our manuscript.

  1. Page 5, line 120, presented higher tissue should be described as a trend.

Thank you for the observation. Corrections were added.

Line 132 - The ACE-2 tissue immunoexpression was increased in the COVID-19 group compared to H1N1 and CONTROL groups (p=0.0005 and p<0.0001, respectively). The COVID-19 group presented higher tissue immunoexpression of IL-33 than the CONTROL group (p=0.013); however, it was not statistically significant compared to the H1N1 group. Regarding CASP-1, the COVID-19 group presented higher tissue immunoexpression than both H1N1 and CONTROL groups (p<0.0001 and p<0.0001, respectively). Respecting tissue immunoexpression of BK receptors and their cleaved forms, the COVID-19 group had a significant increase in B1R (p<0.0001) and B2R tissue expression (p<0.0001) compared with the CONTROL group. Compared with the H1N1 group, there was no statistical significance for both receptors.

Line 171 - Respecting the distribution of MCs in the interstitial and perivascular sites, the COVID-19 group presented a higher number of IgE+ MCs than the CONTROL group (p<0.0001); however, there was no statistical significance when compared with the H1N1 group. The number of Tryptase+ MCs and TB metachromatic MCs were also increased in the COVID-19 group, compared to the H1N1 and CONTROL groups (p=0.039 and p<0.025; p=0.0006 and p<0.0001, respectively). The results can be seen in Table 1 and Figures 1 and 2.                  

  1. Fig 1, the authors can consider revising plots, control, H1N1 and Covid to facilitate easy reading.

Considerations were made regarding the composition of images and captions. The letters of each image related to the groups were exchanged for letters correlating the immunomarker.

Line – 129 Figure 1: Graphics show Comparison between COVID-19 (left images), H1N1 (right upper images) and CONTROL (right lower images), regarding angiotensin-converting enzyme 2, Interleukin-33 and Caspase-1 in the percentage of immunoexpression per HPF (high power field), and density of IgE+ MCs per HPF. (A) ACE-2 has mild positive immunostaining (red arrow) in the alveolar epithelium of the COVID-19 group. However, there is a significant statistical increase compared to the H1N1 and CONTROL groups. (B) Tissue expression of IL-33 (red arrow) appears higher in patients in the COVID-19 group when compared to the CONTROL group; however, it was not statistically increased compared to the H1N1 group. (C) The tissue expression of Caspase-1 (red arrow) is increased in the COVID-19 group compared to the H1N1 and CONTROL groups, demonstrating a more prominent presence of pyroptosis by SARS-CoV-2 infected cells. (D) Regarding the density of IgE+ MCs, the COVID-19 group presented an increase in the number of MCs opsonized by IgE (cells highlighted by red arrows) compared to the CONTROL group. However, there is no statistical difference compared to the H1N1 group. Images were scanned by Axio Scan.Z1 slide scanner (Carl Zeiss, Germany) at 40X magnification. Mann-Whitney nonparametric test. Values ​​of p<0.05 indicated statistical significance.

Line 145 - Figure 2: Graphics show Comparison between COVID-19 (left images), H1N1 (right upper images) and CONTROL (right lower images), regarding the density of Toluidine Blue (TB) MCs, Tryptase+ MCs per HPF (high power field) and tissue bradykinin B1 and B2 receptors in the percentage of immunoexpression per HPF. (A) Metachromatic MCs stained with TB are shown (red arrows) in higher density in the COVID-19 group when compared to the H1N1 and CONTROL groups. (B) The density of Tryptase+ MCs (red arrow) was also increased in the COVID-19 group compared to the H1N1 and CONTROL groups. (C and D) As for tissue immunostaining of bradykinin receptors, B1R and B2R were higher in the COVID-19 group (highlighted by red arrow) than the CONTROL group. However, there was no statistically related difference with the H1N1 group. Images were scanned by Axio Scan.Z1 slide scanner (Carl Zeiss, Germany), B, C and D at 40X magnification and A at 63X magnification. Mann-Whitney nonparametric test. Values ​​of p<0.05 indicated statistical significance.

  1. No data about medication of these patients were mentioned.

Thank you for your consideration. Data on corticosteroids used by patients in the COVID-19 group were included in Table 2.

CHARACTERISTICS

NUMBER OF TRYPTASE+ MCs2

NUMBER OF TOLUIDINE BLUE  METACHROMATIC  MCs2

values

p-valuea

values

p-valueb

TREATED  17/24 (70.8%)1

6.2/2.7 (3 - 11.35)

0.633*

1.2/0.77 (0.35 - 2.45)

0.774*

NON-TREATED  7/24 (29.1%)1

7.7/8.05 (0.7 - 15.25)

1/1.3 (0.55 - 2.75)

CORTICOSTERIDS  ADMINISTERED 3,4

Dexamethasone 6mg/day 12/17

Hydrocortisone 100mg/day 1/17

Hydrocortisone 200mg/day 3/17

Methylprednisolone 125mg/day 2/17

Prednisone 60mg/day 1/17

Prednisone 10mg/day 1/17

Continuous variable by median/interquartile range (minimum-maximum).

1Absolute number (frequency).

2Number of cells per 20 HPF (average).

ap-value compared between TREATED and NON-TREATED patients regarding NUMBER OF TRYPTASE+ MCs.

bp-value compared between TREATED and NON-TREATED patients regarding NUMBER OF TOLUIDINE BLUE MCs.

*Mann-Whitney test.

3Some patients underwent an association with more than one type of drug.

4There is no data related to H1N1 and CONTROL group patients.

  1. The discussion can be shortened. The authors must discuss the results. It looks like  introduction.

Thank you for the suggestion. Arrangements have been made.

  1. It is known that Covid increases the inflammatory parameters, but how mast cells are associated with vascular hyperpermeability are not indicated.

Heparin, among other polyanions by mast cells that are secreted mediators, promotes metachromasia of TB-stained cells. From this, the ability to activate patients and the beginning of the possibility of activating factor XII, whose ability to activate KKS plays a role in the proteolytic cascade that culminates in cases of hyper-COVID-19 vascular capacity. In the present study, COVID-19 and H1N1 group patients had significantly increased results for indicators BR1 and BR2, whose immunoexpression demonstrates inflammatory events and vascular hyperpermeability. As a result of the events mentioned above, intra-alveolar edema can be observed (Table 1).

CHARACTERISTICS

H1N1

COVID-19

CONTROL

values

p-value1

values

p-value2

values

GENDERa

Male

8 (80%)

0.437**

15 (62.5%)

0.709**

8 (72.7%)

Female

2 (20%)

9 (37.5%)

3 (27.3%)

AGEb

43.5 ± 14

0.001***

72 ± 12.5

0.001***

42.3 ± 14.3

TIME FROM HOSPITALIZATION

TO DEATHb

4.7 ± 6.13

0.003***

15.9 ± 10.2

0.051***

7.64 ± 13.1

TIME OF MECHANICAL

VENTILATIONb

4.7 ± 6.13

0.028***

12 ± 9.20

-

-

ALVEOLAR EDEMAa

Present

9 (90%)

0.644**

19 (79.2%)

0.022**

4 (36.4%)

Absent

1 (10%)

5 (20.8%)

7 (63.6%)

TISSUE IMMUNOEXPRESSION OF ACE-2c

3.07/4.26 (1.51 - 9.03)

0.0005*

9.77/13.66 (3.1 - 34.95)

<0.0001*

2.14/3.46 (0.46 - 5.16)

TISSUE IMMUNOEXPRESSION OF IL-33c

10.34/7.51 (1.36 - 19.45)

0.636*

10.14/14.51 (0.52 - 32.75)

0.013*

4.93/3.82 (2.05 - 10.94)

TISSUE IMMUNOEXPRESSION OF CASP-1c

4.57/5,37 (0.36 - 14.34)

<0.0001*

20.56/12.1 (12.23 - 32.85)

<0.0001*

7.73/5.61 (1.09 - 12.67)

TISSUE IMMUNOEXPRESSION OF B1Rc

5.79/7.34 (2.62 - 12.64)

0.636*

6.65/4.45 (1.58 - 12.88)

<0.0001*

1.69/1.65 (0.39 – 5.11)

TISSUE IMMUNOEXPRESSION OF B2Rc

8.29/10 (2.71 - 22.49)

0.249*

13.51/8.87 (3.88 - 23.06)

<0.0001*

1.39/1.62 (0.38 - 6.4)

NUMBER OF IgE+ MCsd

6.97/7.06 (5.55 - 17.1)

0.924*

8.85/3.65 (3.25 - 18.4)

<0.0001*

3.25/1.4 (1.75 - 5)

NUMBER OF TRYPTASE+ MCsd

3.95/4.44 (1.75 - 7.65)

0.039*

6.65/3.4 (0.7 - 15.25)

0.025*

4.65/2.33 (2 - 12.15)

NUMBER OF TB METACHROMATIC MCsd

0.45/0.25 (0.3 - 0.8)

0.0006*

0.85/0.3 (0.6 - 1.4)

<0.0001*

0.6/0.3 (0.4 - 0.9)

Continuous variables expressed by median/interquartile range (minimum - maximum).

aCategorical variables expressed by absolute number (frequency).

bDemographic variable AGE (in years) and clinical variables TIME FROM HOSPITALIZATION TO DEATH (in days) and TIME OF MECHANICAL

VENTILATION (in days) expressed by mean ± straight deviation.

cTissue expression in percentage per HPF.

dNumber of cells per 20 HPF (average).

1p-values compared between COVID-19 and H1N1 groups.

2p-values compared between COVID-19 and CONTROL groups.

*Mann-Whitney test.

**Fisher’s exact test.

***Student’s t-test.

HE stained sample (in “Track Changes”) representing alveolar parenchyma from a patient belonging to the COVID-19 group. Red circles evidence intra-alveolar edema.

  1. Furthermore, role of KKS pathway should be linked clearly to the results and discussion.

We believe that the answer related to the question is found in the paragraph corresponding to line 315. The fact that B1R and B2R receptors are positively immunostained, associated with vascular hyperpermeability and intra-alveolar edema, corroborates the activation and presence of KKS in the patients affected by COVID-19.

Line – 321 In our study, both tissue immunoexpression of B2R and B1R were increased in patients affected by COVID-19 compared to the CONTROL group. The exacerbation of pulmonary vascular permeability and plasma extravasation towards the pulmonary interstitium to the alveolar lumen is evident once intra-alveolar edema was also observed in patients affected by COVID-19 and H1N1 due to the presence of activated bradykinin receptors (Figure 2). These aspects can directly link the plasma leakage events to the KKS activation (Figure 4).

  1. The authors might consider elaborating the results as well.

Thank you for the observation. Corrections were adjusted.

  1. The authors have compared the data to control and H1N1. But no discussion was provided at all.

Thanks for the observation. Data relating to H1N1 has been added to the thread.

Line – 228 In our study, we accounted for many IgE+ MCs in patients who died from COVID-19 compared to the CONTROL group (Table 1, Figure 1) but not when compared to the H1N1 group. Graham et al. demonstrated that MCs have essential participation in inflammatory events caused by the H1N1 Influenza A virus [40]. This large number of IgE+ MCs present in pandemic groups corroborates previous studies.

Round 2

Reviewer 1 Report

Thank you for addressing the comments. The satisfactory changes are made now. 

Author Response

Dear reviewer 1,

Thank you for all your suggestions. Their all were very constructive.

Reviewer 2 Report

To Authors,

The authors have addressed all my concerns and I have no further comments as sees in the responses. Please could you upload a revised manuscript with highlights to facilitate the review process?

Thanks

Author Response

Dear Reviewer 2,

I am sending the manuscript revised and highlighted. 

Thank you for all suggestions. Their all were very constructives.

This manuscript is a resubmission of an earlier submission. The following is a list of the peer review reports and author responses from that submission.